# Assessing the Impact of a Utility Scale Solar Photovoltaic Facility on a Down Gradient Mojave Desert Ecosystem

Dale A. Devitt [1],*, Lorenzo Apodaca [1], Brian Bird [1], John P. Dawyot, Jr. [1], Lynn Fenstermaker [2] and Matthew D. Petrie [1]

1   School of Life Sciences, University of Nevada Las Vegas, Las Vegas, NV 89154, USA
2   Desert Research Institute, Las Vegas, NV 89119, USA
*   Correspondence: dale.devitt@unlv.edu

**Abstract:** A field study was conducted in the Mojave Desert (USA) to assess the influence of a large photo voltaic facility on heat and water transport into an adjacent creosote (*Larrea tridentata*) bursage (*Ambrosia dumosa*) plant community. Air temperature, plant physiological status, soil water in storage and precipitation were monitored over a two to four year period. A service road built 27 years before the construction of the PV facility decoupled the wash system at the site leading to a significant decline in soil moisture, canopy level NDVI values and mid-day leaf xylem water potentials ($p < 0.001$) down gradient from the PV facility. Measurements along a 900 m gradient suggested that plants closer to where the wash was decoupled were placed under significantly greater stress during the higher environmental demand summer months. Air temperatures measured at three 10 m meteorological towers revealed warmer night time temperatures at the two towers located in close association with the solar facility (Inside Facility—IF and Adjacent to facility—AF), compared to the Down Gradient Control tower (DGC). As the warmer air was displaced down gradient, the temperature front advanced into the creosote—bursage plant community with values 5 to 8 °C warmer along an east west front just north of tower AF. Based on our research in Eldorado Valley, NV, USA, a down gradient zone of about 300 m was impacted to the greatest extent (water and heat), suggesting that the spacing between solar facilities will be a critical factor in terms of preserving high quality habitat for the desert tortoise and other species of concern. Greater research is needed to identify habitat zones acceptable for animal populations (especially the desert tortoise) within areas of high solar energy development and this should be done prior to any fragmentation of the ecosystem.

**Keywords:** plant stress; precipitation; soil water content; NDVI





## 1. Introduction

As the United States seeks further energy independence, many states are moving toward greater diversification of their energy portfolios. In the case of Nevada (USA), a goal has been set to meet 50% of the energy demand by 2030 with renewable energy [1]. As such, large tracts of land are being approved or are in the approval process for the installation of solar photovoltaic facilities. In 2011 the Bureau of Land Management (BLM) identified about 9 million ha of federal land in the west that met strict criteria for solar development [2] with the major limitation on development controlled primarily by the sites geographical potential [3]. New facilities in Nevada that have been given approval to move forward are as great as 2900 ha in size (2022, Primergy Solar, 440 MW, 25 miles north of Las Vegas, NV, USA). Solar energy is a clean energy [4–6], but it does have significant unintended impacts on desert ecosystems by altering surface hydrology [7,8], energy balances and surface air temperatures [9–11], biodiversity and ecosystem services [12–14] and causing habitat fragmentation [15–17]. Deserts are known to be fragile ecosystems with recovery from disturbances predicted to take from decades to centuries [18]. In southern

Nevada, the Mojave Desert is home to the threatened desert tortoise [19] so it is critical that scientists understand the impact solar facilities have on desert ecosystems. Although it should also be noted that PV facilities can have a positive impact on plants and animals and surface micro habitats within the facilities [13,19] and that recent climate models associated with large scale solar development in the Sahara Desert predict increased coverage of vegetation creating a positive feedback that would further increase rainfall [11].

In the Mojave Desert, the valley floors are dominated by creosote (*Larrea tridentata*) bursage (*Ambrosia dumosa*) plant communities and although precipitation is low (long-term average for Las Vegas, NV, USA is slightly less than 11 cm per year), [20] extensive shallow wash systems often allow for rainwater harvesting [7,21,22]; the moving of additional water (surface runoff) to down gradient locations. Popcewicz et al. [23] predicted that the greatest impact by energy development in western North America will be on shrublands; such as creosote bursage plant communities found throughout the Mojave Desert.

Any structure that has a large footprint can alter water movement in a desert wash system. Construction of roads, transmission lines and utility scale solar photovoltaic facilities can decouple up-gradient washes from down-gradient locations [21,22], leading to a decline in soil water in storage. Whether such water deficit conditions translate into elevated levels of plant stress would depend on how close the plants are to the decoupling zone, the drought tolerance of the species and whether the plants are located in close proximity to a wash.

Photovoltaic facilities can also alter the energy balance by generating sensible heat in the process of capturing electromagnetic energy by solar panels [10]. Some of this extra heat that is subsequently released is displaced into adjacent lands [9]. It is estimated that approximately 63% of incoming solar is transmitted through the panels [24] with Wynne [25] reporting the underside of solar panels as high as 10 °C higher than the absorbing face of the panels during the day. As the size of solar photovoltaic facilities become larger and join up with adjacent facilities to create centers of solar development such as in Eldorado Valley, NV, USA (several million panels), providing corridors for animal movement will become critical. As such, designated conservation land must not only exist but also not be significantly altered that overall health of the plant community declines leading to altered ecological flow between patches [26]. Highly fragmented areas may impact a wide range of species that lack behavioral skills to traverse such areas [27,28]. Andrews [27] argued that it is far more expensive to maintain unviable habitats for threatened species than to simply leave viable areas undisturbed when such options are still available.

Our research was undertaken to help close the gap in knowledge on the impact/conflict utility scale solar photovoltaic facilities (1 km$^2$ in area) have on ecosystems [12,29,30]. As such we assessed the impact a large PV facility located in the Mojave Desert had on an adjacent down gradient creosote bursage plant community. We monitored changes in soil and plant water status over a 900 m transect, compared micro climates, assessed water and energy status based on data from 10 m meteorological towers separated by as much as 2.3 km and assessed heat movement into the plant community. We hypothesized that plants growing down gradient but closer to the solar facility would be under greater stress because of altered soil moisture associated with the hydrologic decoupling of washes and elevated ambient temperatures associated with the generation of sensible heat and night time longwave radiation.

## 2. Material and Methods

Research was conducted at a field site in Eldorado Valley, NV, USA associated with the Copper Mountain solar facility; within the Boulder City Energy Resource Zone. The photovoltaic facility was built in two phases with "phase I" completed in September 2012 and "phase two" completed in October 2015 (Figure 1). The facility is approximately 180 ha in size. To the west (~0.5 km) of the Copper Mountain solar facility is an additional facility that is approximately 120 ha in size as well as a third facility to the northwest that is also

approximately 120 ha in size and a fourth facility to the west of the third facility that is 113 ha in size. To the south of the Copper Mountain facility is approximately 7 km of undeveloped land with a downward slope of 4.5% toward the solar facilities. The panels are in a fixed array, with 30 degree panel slopes facing due south. Four panels 120 cm × 60 cm with 5 cm spacing's are set in vertical arrays (10 × 10) that run horizontally for as much as 147 m. The horizontal arrays are separated by access paths that are approximately 4.5 m between vertical support beams. The top of the panels are approximately 160 cm off of the soil surface while the bottom of the panels are approximately 41 cm off of the soil surface.

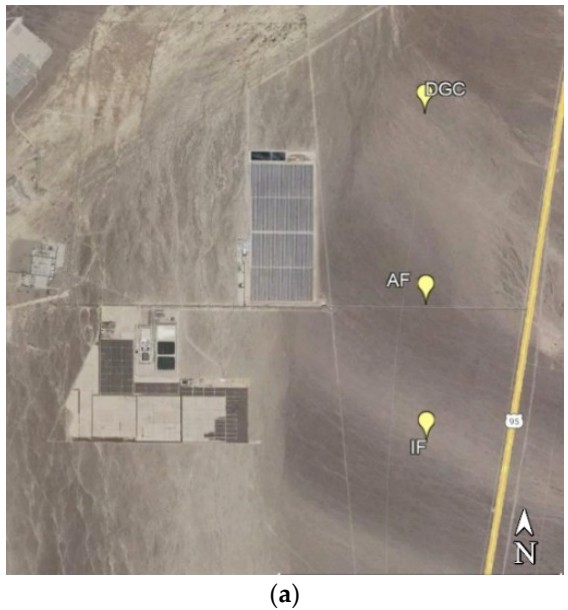 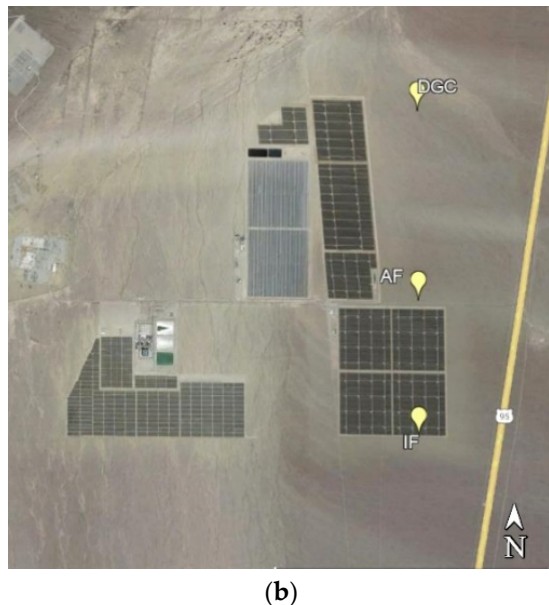

| (**a**) | (**b**) |

**Figure 1.** Site location map. Google Earth images of the study area in 2010 (**a**) and 2015 (**b**). 10 m meteorological towers were placed at the site in 2015 and are indicated with yellow thumbtacks; IF = Inside facility, AF = Adjacent to Facility and DGC = Down gradient Control.

The solar facility is situated in a well-established creosote-bursage plant community (site location map). A 4% downward slope from south to north exists within the monitoring area with a network of shallow washes. Three monitoring sites were established with each containing a 10 m meteorological tower; one located approximately 11 m inside the south fence of the Copper Mountain Solar facility (IF: Inside facility, 35°46′44.41″ N, 114°57′29.95″ W, elev. 615 m), situated on packed and modified bare soil within 12 m of the first row of solar panels. A second tower (AF: Adjacent to Facility, 35°47′36.17″ N, 114°57′29.17″ W, elev. 582 m) was located within the creosote bursage plant community approximately 36 m north of an asphalt service road (7.7 m wide) that runs east west of the solar facility and approximately 100 m from the northern most row of solar panels. This service road is visible from Google Earth images (Mountain View, CA, USA) acquired in 1985 as a dirt road, however by 1990 the road appears to be asphalt covered, representing a 29 to 34 year period (2019) in which the road decoupled up gradient washes from down gradient washes. The third tower (DGC: Down-gradient Control, 35°48′49.77″ N, 114°57′28.46″ W, elev. 535 m) was located approximately 2.3 km to the north of tower AF also in a mature creosote bursage plant community. Soil at all three sites was classified as a Hypoint gravelly sandy loam, with a 0–4% slope [31].

The 10 m towers were equipped with multiple sensors at heights that varied based on the sensor. Sensor heights were at 1 m, 2 m, 6 m and 10 m. These sensors included a net radiometer at 6 m (Kip and Zonen CNR1, Delft, The Netherlands), a Hydro Svs TB4 tipping bucket at 2 m, and a Quantum sensor (LiCor 1905A, Lincoln, NB, USA) at 2 m. Propeller anemometers were positioned at 1 and 6 m (wind vane RM Young 05103, Traverse City, MI, USA) while sonic anemometers (Gill Instruments, Lymington Hampshire UK) were

positioned at 2 and 10 m. Barometers (Setra 278, Boxborough, MA, USA) were located at 2 m. Capacitative RH sensors (CSI HMP50 temperature and relative humidity, Campbell Scientific, Logan, UT, USA) were positioned at 1 m, 2 m, 6 m and 10 m. Constantan thermocouples were buried at depths of 1, 5, 10, 20 and 40 cm (Omega Engineering, Norwalk, CT, USA). Soil heat flux plates (Hukseflux HFP015C, Center Moriches, NY, USA) were positioned at a soil depth of 8 cm. Time domain reflectometer sensors (CSI C5616, Campbell Scientific, Logan, UT, USA) were placed at soil depths of 5, 10, 20 and 40 cm. Pyranometers (LiCor 200SZ, Lincoln, NB, USA) were placed at 2m. Normalized Difference Vegetation Index sensors (Skye Instruments Ltd., Llandrindod Wells, UK) were located at a height of 40 cm above plant canopies at towers AF and DGC with data processed following the approach of Devitt et al. [32].

At towers AF and DGC we took intact soil cores (0–10 cm) to measure saturated hydraulic conductivity back in the laboratory [33]. We were unable to successfully drive the brass rings into the soil at AF (compact soil and gravel). We also measured infiltration and sorptivity near towers AF and DGC using mini infiltrometers (Meter Group, Pullman, WA, USA)

A 900 m soil plant transect was established with monitoring locations situated every 100 m, starting at a site adjacent to tower AF. Each site had an access tube installed to a depth of 100 cm which allowed for a theta probe (PR2, Delta-2, Cambridge, UK) to be lowered into the access tube to estimate soil volumetric water content at depths of 10, 20, 30, 40, 60 and 100 cm. On a monthly basis, mid-day leaf xylem water potential was assessed with a pressure bomb (PMS Instruments, Albany, OR, USA), Canopy temperatures with an infrared thermometer (39800 Infrared Thermometer, Cole Palmer, Vernon Hills, IL, USA) and chlorophyll index with a chlorophyll index meter (Field Scout CM1000 Chlorophyll Meter, Spectrum Technologies, Aurora, IL, USA).

Air temperatures were also measured with ibutton sensors (Maxim Intergrated, San Jose, CA, USA) mounted in ventilated chambers at heights of 10 cm, 1 m, 2 m and 3 m on vertical poles. The poles were positioned in a grid pattern (six N/S by four W/E arrangement) spaced 300 m apart, starting at a position 432 m to the west of tower AF. Detailed measurements were taken on 30 May 2015 and 1 September 2015.

Satellite NDVI values were obtained from Landsat. Values were extracted from 1991 (prior to solar development and in 2015 after solar development at AF and DGC (not at IF which was bare soil). Landsat NDVI data were acquired for the meteorological tower locations using GPS coordinates. Although other vegetation indices (such as the soil adjusted vegetation index—SAVI [34] and modified SAVI—MSAVI [35] might be more applicable for an area with a high percentage of bare soil, the NDVI was acquired to provide a better comparison with the ground-based NDVI measurements (sensors and meter). The NDVI data for the tower sites were exported from the Climate Engine website (climateengine.org; [36]). Climate Engine was developed by faculty from the Desert Research Institute and University of Idaho using Google Earth Engine (Mountain View, CA, USA). This website provides consistent time series Landsat NDVI data for specific locations and ensures that all data (regardless of year and Landsat satellite) are processed in a uniform manner enabling reliable comparison of NDVI values over time. Percent vegetative cover was also assessed from Google Earth imagery for an approximate area of 152 m by 107 m immediately north of towers AF and DGC, with the earliest and latest dates with comparable spatial resolution. In this case, May 2010 and March 2022. The extracted images were loaded into ArcGIS where an iso cluster unsupervised classification was performed to differentiate vegetation from soil. The pixel counts for the resulting bare soil and vegetation in each classified image were used to calculate percent cover.

Soil and plant measurements along the 900 m gradient only occurred during 2015 and 2016 whereas all measurements associated with the three towers occurred from 2015–2019.Data was processed using Sigma Stat and graphs generated using SigmaPlot (Systat Software Inc. Point Richmond, CA, USA). Statistical analysis included descriptive statistics, Analysis of Variance -and linear and multiple regression analyses to assess soil, plant, water

and atmospheric interactions. Multiple regressions were generated in a backward stepwise manner, with elimination of terms occurring when *p* values for the *t*-test exceeded 0.05. To prevent the possibility of co-correlation, parameters were only included if variance inflation factors (VIF) were ≤2.0 and the sum total of VIF's was ≤10.0. If these threshold VIF's were exceeded, parameters were eliminated and regression analyses were rerun until acceptable VIF's were attained.

## 3. Results

### 3.1. Site Conditions

The experiment was conducted in a north south oriented valley. Wind was primarily from the south (53 +/− 6% of hourly values from 135° to 225°) during spring and summer months dropping off to only 28 +/− 8% of hourly values during fall winter months. All of the monitoring sites fell within the same soil classification (Hypoint, loamy sand) with saturated hydraulic conductivities measured on intact soil cores (0–10 cm) at towers AF and DGC (tower IF excluded because of heavy compaction) revealing no statistically significant differences (4.1 +/− 4.4 cm hr$^{-1}$ vs. 3.5 +/− 3.6 cm hr$^{-1}$, *p* = 0.70) nor were there any statistical differences in sorptivities (0.16 cm sec$^{-0.5}$ +/− 0.14 vs. 0.15 cm sec$^{-0.5}$ +/− 0.09. *p* = 0.91) measured with mini infiltrometers.

Precipitation averaged 12.3 cm per year (+/−6.4) over the extended 57 month period (Table 1) with yearly ET$_{ref}$ measured during the period of 2016–2019 averaging 183 +/− 27 cm at tower IF and 201 +/− 18 cm at tower AF (*p* > 0.05, no estimates available from tower DGC). Precipitation at towers AF and DGC (monthly totals) were highly correlated (R$^2$ = 0.89, *p* < 0.001) with a 57-month precipitation total identical at towers AF and DGC (51.6 cm which included 9 months in 2015) compared to 66.3 cm at tower IF which was located on the south side of the solar facility. Precipitation totals in 2016 accounted for the majority of this difference with tower IF precipitation approximately 15 cm higher than tower AF and approximately 13 cm higher than tower DGC. These higher precipitation values in 2016 were primarily associated with storms from the south east (high values; wind direction; 127 +/− 58° vs. low values; 322 +/− 59° *p* = 0.01) with higher wind speeds (high values, wind speed 5.2 +/− 1.8 ms$^{-1}$ vs. low values, 2.5 +/− 0.9 ms$^{-1}$ *p* < 0.05).

**Table 1.** Yearly Precipitation (PPT) and ET$_{ref}$ totals in cm based on tower location.

| Year | Tower IF | | Tower AF | | Tower DGC |
|------|----------|--|----------|--|-----------|
| | PPT (cm) | ET$_{ref}$ (cm) | PPT (cm) | ET$_{ref}$ (cm) | PPT (cm) |
| 2016 | 21.3 | 184.5 | 6.3 | 223.1 | 8.6 |
| 2017 | 8.2 | 201.7 | 7.2 | 196.6 | 5.9 |
| 2018 | 9.0 | 201.5 | 8.7 | 203.0 | 11.1 |
| 2019 | 22.8 | - | 21.1 | - | 18.2 |

Towers AF and DGC had similar soil types and identical long-term precipitation totals which led to no significant differences in the soil volumetric water contents at any depth (Figure 2, *p* > 0.05). In particular, at the 40 cm depth we observed little response in the soil water content over time. However, using a boxplot does not provide one with an assessment of the underlying distribution of the data in each group nor the observation number. As such, we provide the raw data (Figure 3) to give greater insight into the episodic response of the soil volumetric water content to the low rainfall desert environment at our study site. We examined 17 high rainfall events (>1 cm/day) occurring at towers AF and DGC which revealed no significant differences (*p* > 0.05) in the median values of rainfall intensity nor in the median rainfall amounts, however we did find a significant difference (*p* < 0.001) in the soil volumetric water content at the 5 cm depth (0.184 tower AF vs. 0.254 tower DGC) reflecting a 38% higher average value at the DGC tower. We also assessed the hourly values greater than 0.10 (a value always associated with rainfall events). We ran a

Mann Whitney rank sum test comparing the values greater than 0.10 at towers AF and DGC. The values greater than 0.10 were found to be significantly different at the $p < 0.001$ level at all depths except the 40 cm depth ($p > 0.05$). A clearer contrast was made by summing up the soil volumetric water contents above the baseline 0.10 value (Tower DCG at 5, 10, 20 and 40 cm; 865, 667, 678 and 136 compared to Tower AF; 272, 341, 110, and 85). Sixteen percent of all soil volumetric water content values at the 5 cm depth measured over the 4.5-year period at tower DCG were above 0.10 compared to just 6% at tower AF, dropping to 3 vs. 2% at the 40 cm depth.

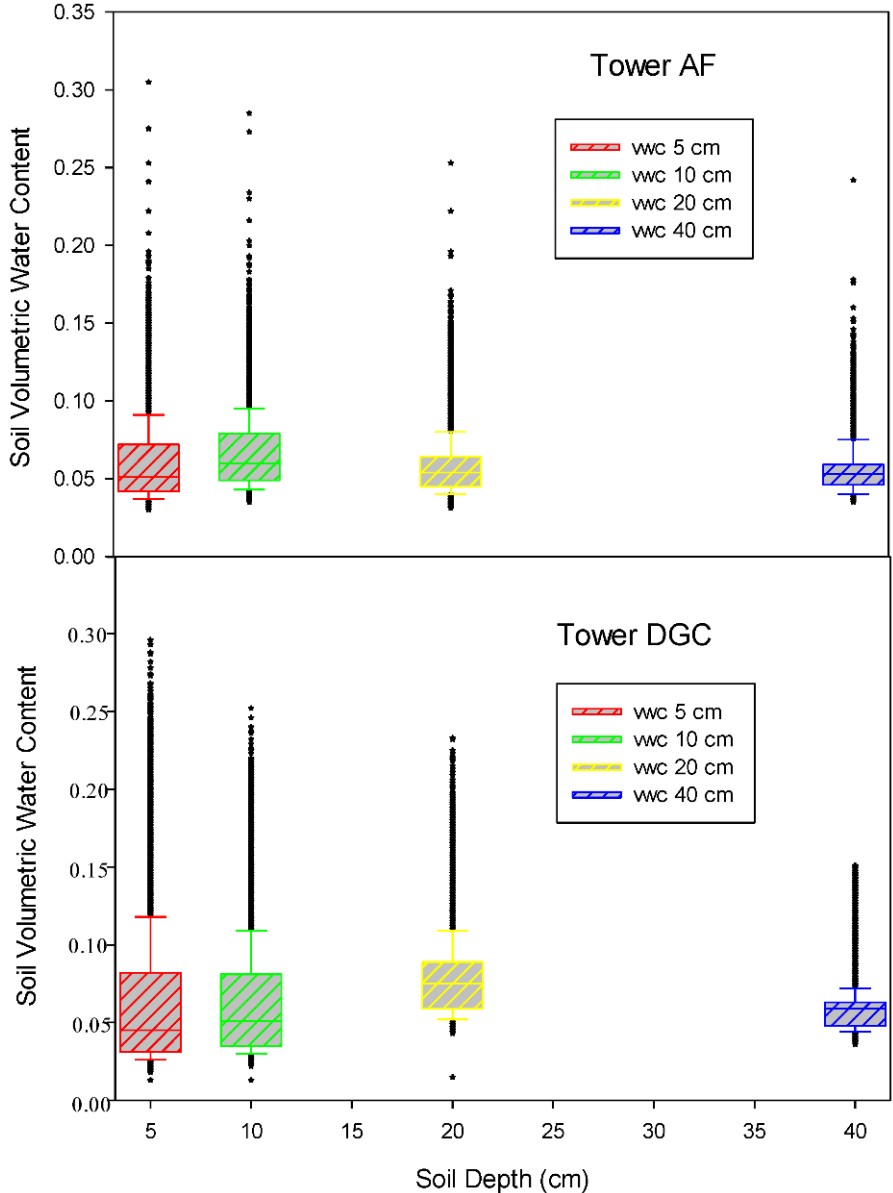

**Figure 2.** Boxplots of soil volumetric water content at the 5, 10, 20 and 40 cm depths for both the AF and DGC tower locations. The boxplots reveal the median value, upper and lower quartiles, minimum and maximum values and potential outliers. The median values revealed no significant differences in soil volumetric water content with depth or between sites.

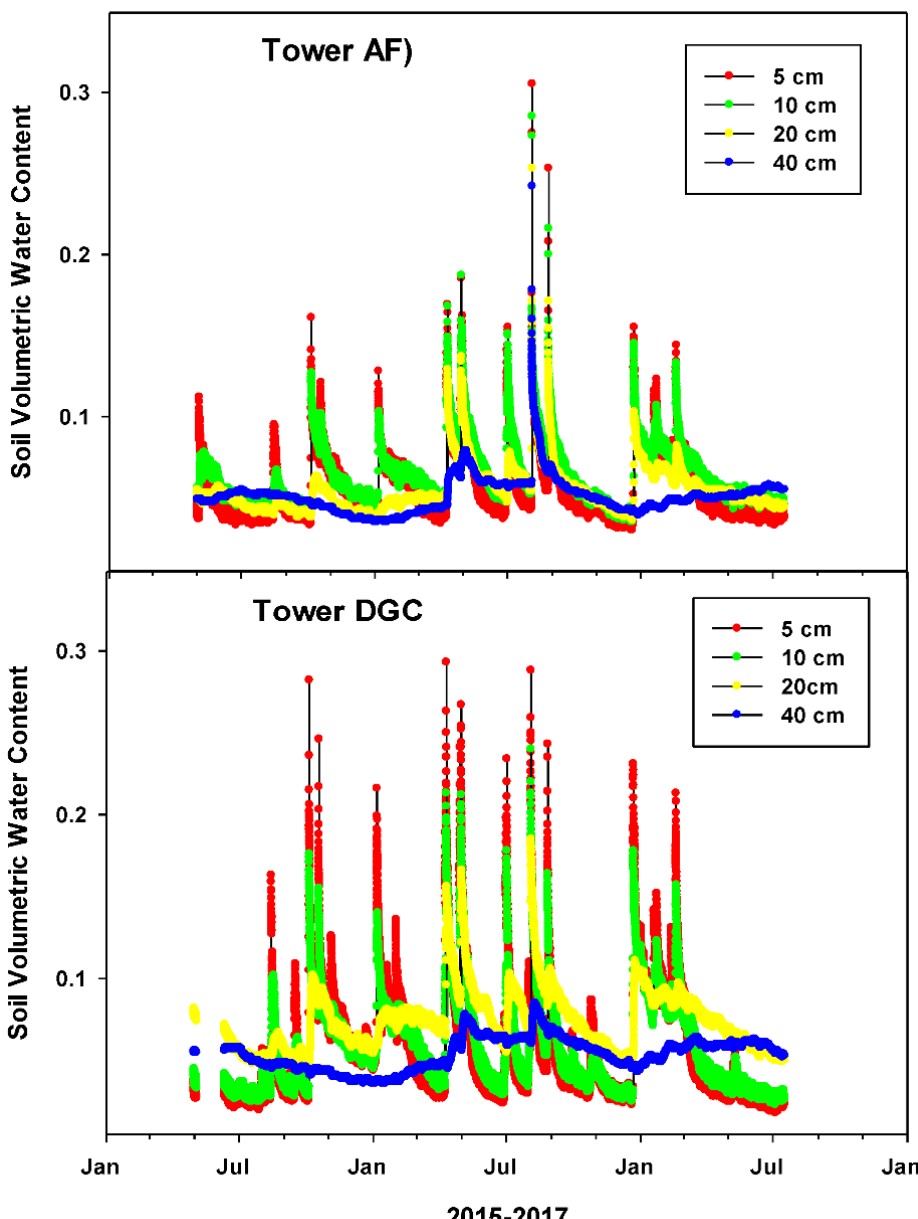

**Figure 3.** Soil volumetric water content at depths of 5, 10, 20 and 40 cm at tower AF and DCG during the monitoring period, revealing the episodic nature of the soil volumetric water content.

Because the saturated hydraulic conductivities and infiltration rates were not significantly different between the two sites to the north of the solar facility, the data suggested that additional water via runoff from the up gradient area between towers AF and DGC was probably contributing additional water to the area near the DGC tower. Both towers AF and DGC had a similar number of soil volumetric water content peaks at the 5 cm depth (28 peaks at tower AF vs. 32 peaks at tower DGC), however only 2 of the peaks at tower AF exceeded a soil volumetric water content of 0.20 whereas at tower DGC 21 peaks exceeded 0.20. The number of days the soil volumetric water content at the 5 cm depth was between values of 0.15 and 0.19, 0.20 and 0.24 and $\geq 0.25$ was 100, 38 and 18 days at tower DGC vs. 25, 4 and 1 days at tower AF.

Both towers AF and DGC revealed highly correlated linear relationships between monthly precipitation and monthly changes in soil water in storage (0–50 cm depth, $R^2 = 0.70$, $p < 0.001$ tower AF vs. $R^2 = 0.87$, $p < 0.001$ at tower DGC) indicating that at a given monthly precipitation total, greater storage change would be predicted at tower

DGC ($p < 0.001$), such as a 4 cm precipitation total driving a soil water storage change of 8.1 cm at tower AF but 9.1 cm at tower DGC. We assessed soil water in storage changes which revealed an 18% higher cumulative storage change at tower DGC compared to tower AF suggesting a possible runoff contribution value to apply to the tower DGC area. Based on soil water in storage changes at the ten monitoring sites located along the 900 m gradient starting at tower AF, little change was noted for locations outside of the washes however, within the washes storage changes responded more dramatically to heavy precipitation especially at the farthest location (900 m) down gradient from tower AF (Figure 4). Average soil water in storage at the 10 monitoring locations over time revealed that 7 out of 10 dates in 2015–2016 were statistically higher ($p < 0.05$) in the wash compared to outside of the wash. It should be noted that at tower AF, soil water storage change revealed no response in or out of washes whereas at a undisturbed site S of the facility, the greatest storage change (>8 cm after a heavy rainfall event in October 2015) was observed indicating that disturbance had a clear decoupling effect on the wash system (2 cm change at the AF tower after the same rainfall event).

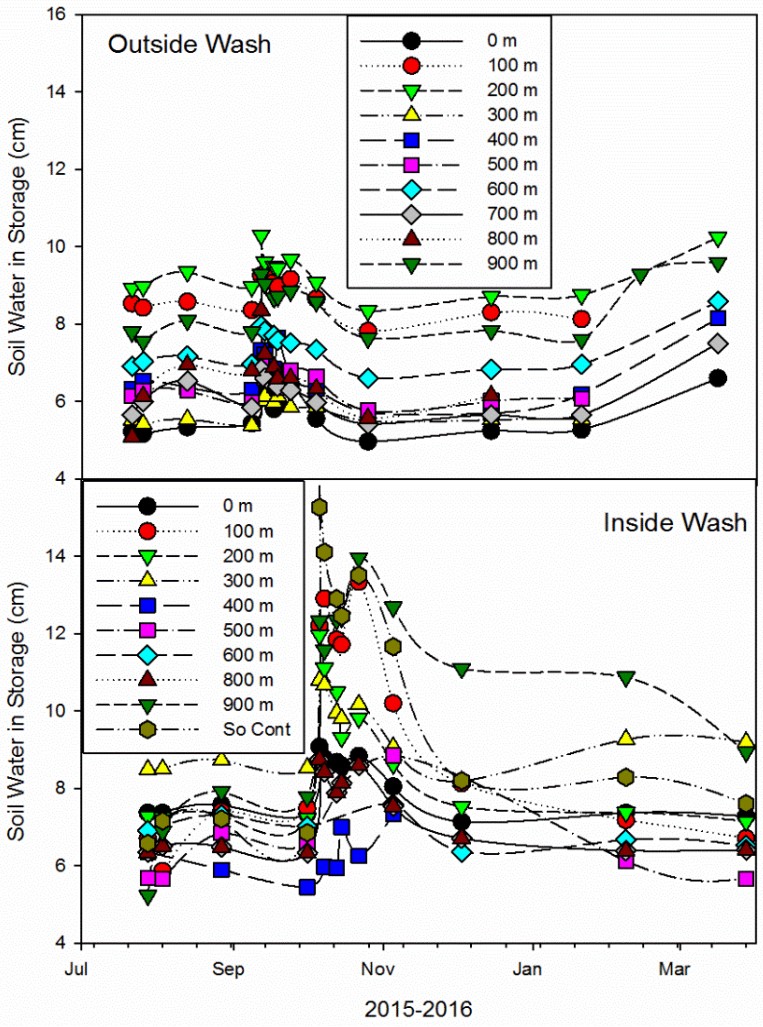

**Figure 4.** Soil water in storage (cm) estimated along the 900 m gradient inside and outside of the wash. Also included are estimates at a control site within the wash system upgradient and south of the solar facility (South Control). During seven of the ten dates in 2015–2016, the average soil water in storage was higher in the wash than outside of the wash ($p < 0.05$).

*3.2. Plant Response*

Change in stem elongation, shrub height and canopy volumes of creosote were not significantly different ($p > 0.05$) based on plants growing inside or outside of the washes or at towers AF and DGC. However, seed production was shown to be significantly higher from plants growing outside of the washes (11.2 +/− 1.1 seeds per 20 cm stem cuttings) compared to plants growing inside of the washes (7.3 +/− 1.3 seeds).

Leaf xylem water potential of creosote measured at mid-day revealed a two phase linear fit (slope-plateau) over the 900 m gradient starting at tower AF (Figure 5). Plants located within the first 300 m of tower AF had significantly ($p < 0.001$) lower leaf xylem water potentials (−5.7 +/− 1.4 MPa) compared to plants growing at locations from 400–900 m down gradient (−4.4 +/− 0.8 MPa). However, these lower leaf xylem water potentials values during the summer period (closer to tower AF) became non distinguishable from all other values obtained along the 900 m gradient during a wetter non stress period (February 2016, all sites, −3.3 +/− 0.2 MPa).

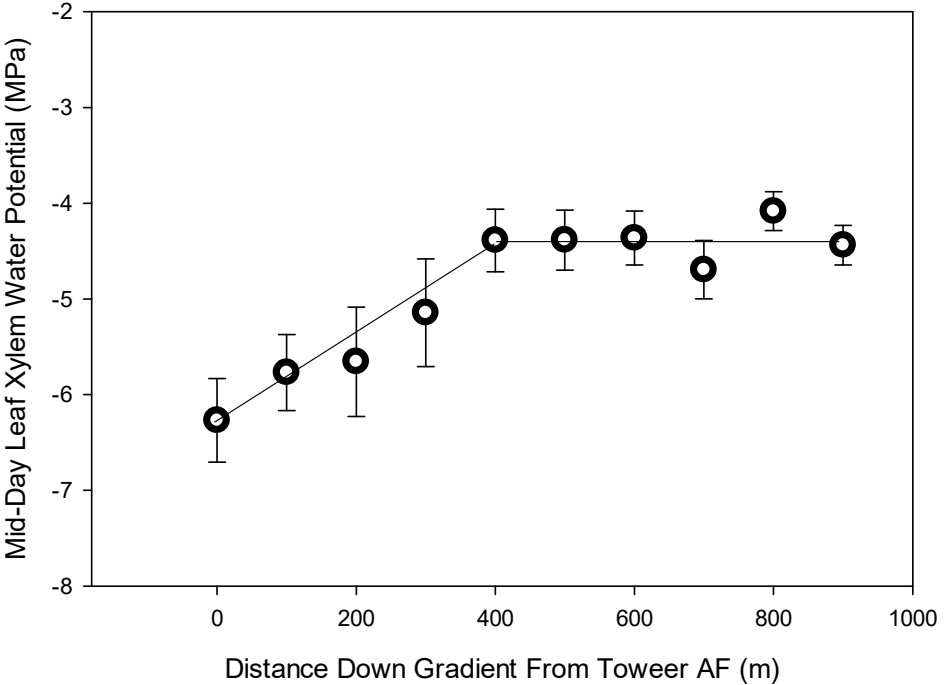

**Figure 5.** Creosote leaf xylem water potential with error bars measured at mid-day along the 900 m gradient starting at tower AF during June–October 2016.

Canopy minus ambient temperatures (delta T) of creosote taken at mid-day during summer and winter varied along the 900 m gradient but were not statistically different (−1.82 summer vs. −1.88 winter. $p = 0.48$). During the summer of 2015 only 3 of the 10 delta T values were positive while during the winter only 2 of the 10 values were positive. However, 4 of the 5 positive values occurred at either the zero location (Tower AF) or at 100 m from Tower AF, with the highest delta T value approaching 2 °C. The fact that 15 of the 20 delta T values were negative during summer and winter suggested that creosote was able to maintain low delta T values through evaporative cooling via transpiration.

Chlorophyll index of creosote (based on red and NIR) was taken at mid-day on a monthly basis whereas continuous measurements of NDVI occurred with insitu sensors placed over the canopies at towers AF and DGC. Cumulative delta chlorophyll index values generated for plants growing at 900 m vs. at 0 m inside and outside of the wash are reported in Figure 6. Outside of the wash, chlorophyll index was almost always higher at the 900 m monitoring location compared to the 0 m location during 2015–2016 whereas

inside the wash there was a steeper gradient over the first 7 months of monitoring covering the wetter lower environmental demand period. However, following this less stressful period, creosote growing inside the wash revealed very little difference in chlorophyll index between the 0 and 900 m locations.

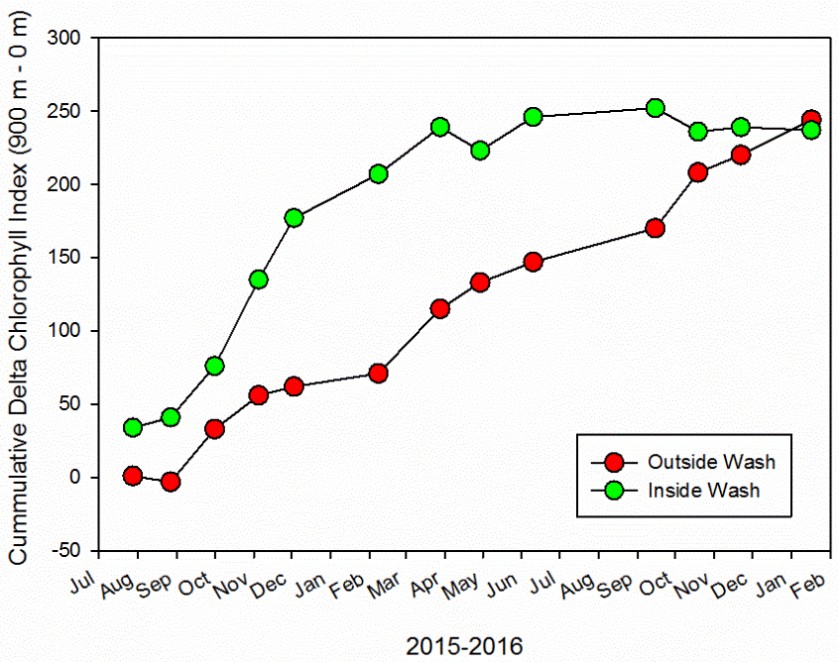

**Figure 6.** Cumulative delta chlorophyll index values measured on creosote plants at the end of the 900 m gradient minus values at the beginning of the 900 m gradient (next to tower AF), significantly different at the *p* < 0.001 level.

NDVI measured on the canopies of creosote (insitu sensors, Figure 7) at towers AF and DGC reveled significant differences (*p* < 0.001) in the median values acquired between mid-October to mid-December and also from mid-December through mid-March. Greater separation occurred during the latter period with some values almost twice as high at the DGC location, suggesting differences in water availability and levels of plant water stress. NDVI assessed with Landsat revealed no correlations with canopy level NDVI values (*p* > 0.05) but did reveal significant differences in NDVI values at towers AF and DGC (*p* = 0.014) but with higher median values at tower AF (0.0986) than at tower DGC (0.0877) which we believe was directly associated with low sensor sensitivity in the sparse vegetated landscape, suggesting a soil adjusted vegetation index might be a better approach when working at the larger scale. Plant canopy NDVI values with sensors were as high as 0.60 whereas satellite values integrated over the sparse vegetation were typically less than 0.10. At the smaller scale, we believe the chlorophyll index meter and NDVI sensors detected subtle but significant changes that Landsat NDVI could not detect, which would mean great care must be taken in evaluating plant response when using satellite NDVI values [37]. and that including NDVI sensors with satellite data acquisition would be a good approach to assess the sensitivity of the data acquired from satellites [32].

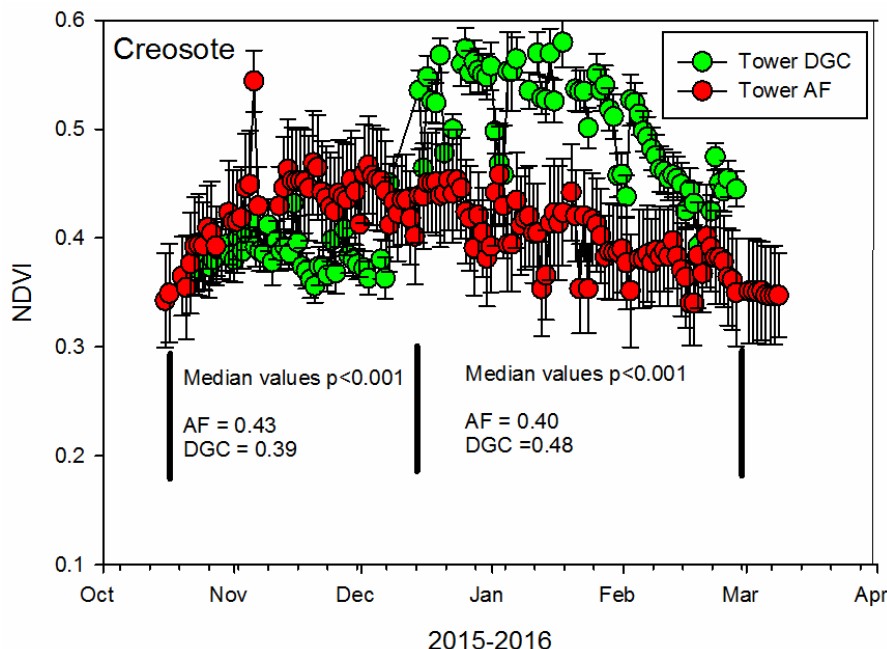

**Figure 7.** NDVI (overhead sensors) of creosote canopies near tower DGC and AF over a 5-month period with significant ($p < 0.001$) separation during the late fall/winter and late winter/early spring periods in 2015–2016.

Percent vegetative cover was assessed using Google Earth images in 2010, five years before completion of the Copper Mountain solar facility and then in 2022, seven years after completion, revealing nearly identical tower location values in 2010 (Tower AF 17.3% vs. Tower DGC 17.8%) and in 2022 (Tower AF 13.3% vs. Tower DGC 13.8%). A percent decline of 23% at tower AF and a 22% decline at tower DGC we believe is significant but the fact that the control site declined in a similar fashion suggested that other factors than closeness to a solar facility influenced the loss in cover over this 12-year period.

*3.3. Air Temperature*

Monthly hourly average air temperatures (Figure 8) for the months of December (greatest differences), March and August demonstrate the seasonal change at the 1 m height, revealing significantly warmer air temperatures ($p < 0.001$) at the two towers near the solar facility compared to tower DGC which was 2.3 km down gradient from tower AF. We recognize that with a 4% slope some cold air drainage may have been occurring, however, following such logic cold air drainage may also have been occurring through the solar facility from up-gradient locations further to the south. Differences were greatest during nighttime hours with no significant differences ($p > 0.05$) during peak sunshine hours. However, at the 2 m height at tower DGC (data not shown) higher temperatures were recorded during the sunshine hours suggesting a possible cold air displacement associated with transpiration from the creosote plants enhancing convection of warm air from below the plants which had canopy heights of approximately 1 m. Only during the month of December were hourly temperatures at 1 m recorded below freezing and only at tower DGC, whereas near the solar facility nighttime temperatures always remained above freezing. In Figure 9 we report the total monthly degree hours that were warmer at tower AF than at tower DGC. November through March all had higher average monthly degree hours (1154 +/− 179 monthly average degree hours) warmer at tower AF compared to tower DGC. November through March contrasted with April through October which averaged only 524 +/− 73 degree hours (significantly different, $p < 0.001$). The greatest temperature difference occurred at 8 AM at the 1 m height in December when the temperature at tower AF was 4.1 °C warmer than at tower DGC.

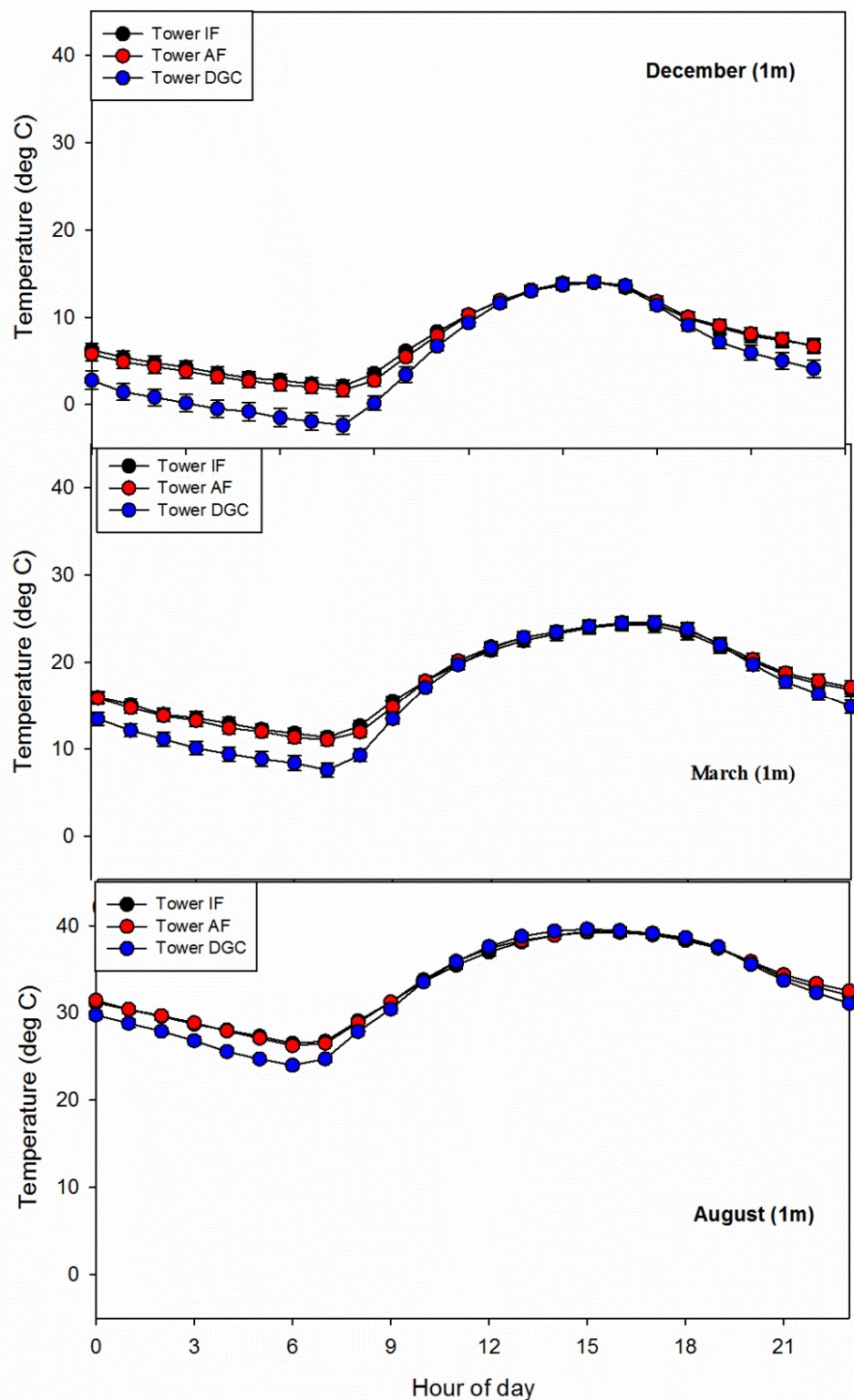

**Figure 8.** Air temperature hourly average plus error bars for the months of December, March and August 2016 at a height of 1 m at towers IF, AF and DGC.

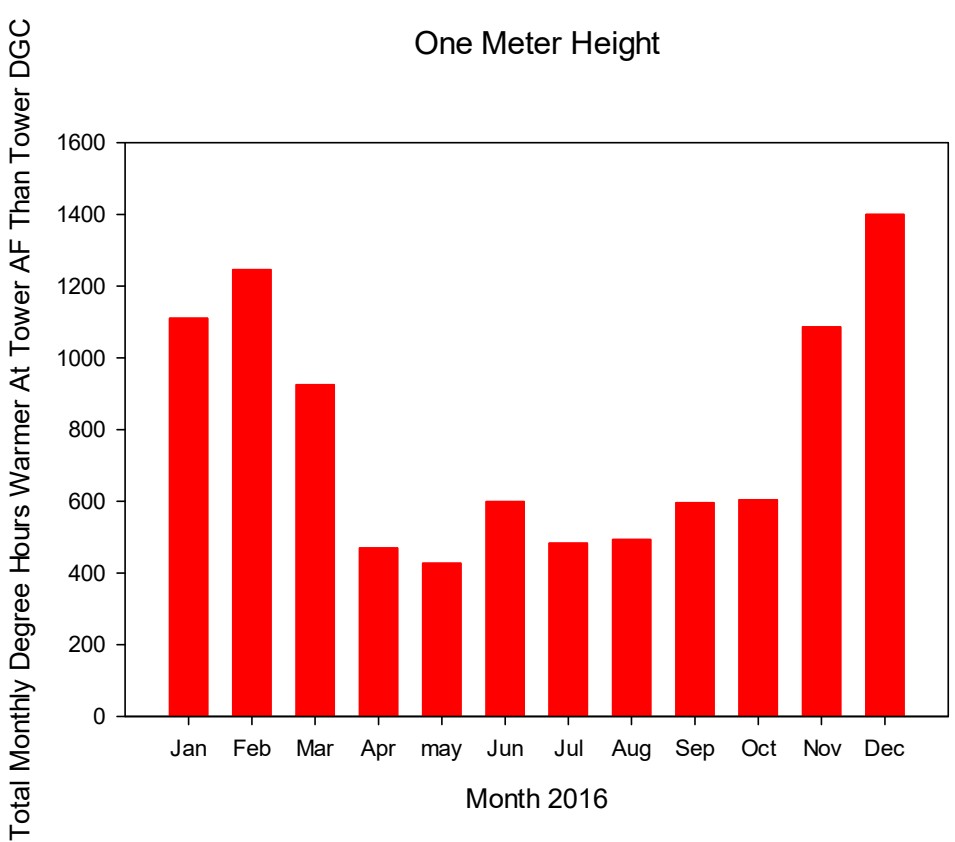

**Figure 9.** Total monthly degree hours warmer at tower AF than tower DGC in 2016.

### 3.4. Spatial Temperature Patterns

I-button temperature sensors placed at four heights in a 6 by 4 grid spaced 300 m apart allowed for temperature contour maps to be generated. In Figures 10 and 11 we report data for all four heights for 30 May and 1 September 2015. These maps revealed warmer temperatures nearer to the southern edge of the grid, which was closest to the solar facility, with greater warm air displacement to the north at the higher monitoring heights. Temperatures measured during May revealed values as much as 6 °C warmer at the south edge compared to a region in the NE part of the grid (~1100 m north along the east boundary), whereas at the 200 cm height on 1 September, temperatures were as great as 8 °C warmer nearer to the south edge compared to the north edge. However, at the 300 cm height the entire area recorded little differences in air temperatures, with all values very close to 31 °C. Although the temperatures were significantly warmer during September than in May, the area mapped at the highest temperature interval was very similar, revealing larger warmer areas at higher heights, with very similar values at the 1 m height of the creosote (40% area in May vs. 39% in September) while at 300 cm 87% of the area was mapped at the higher temperature interval in May vs. 100% in September.

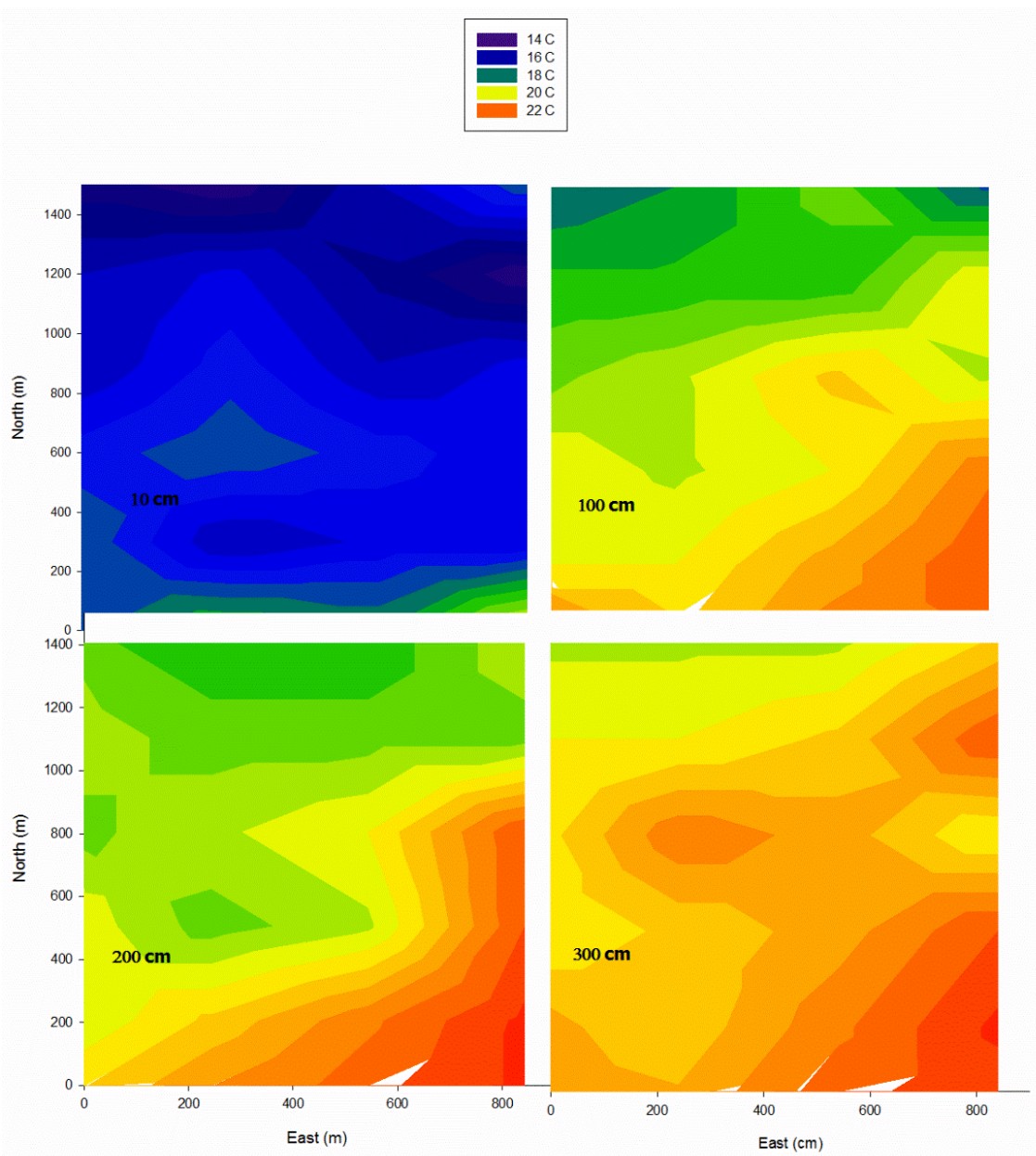

**Figure 10.** Air temperature at 0700 hour on 30 May at heights of 10, 100, 200 and 300 cm within the down gradient creosote bursage plant community. The solar facility was located approximately 100 m up gradient from the lower west to east edge of the monitoring area (see site location map).

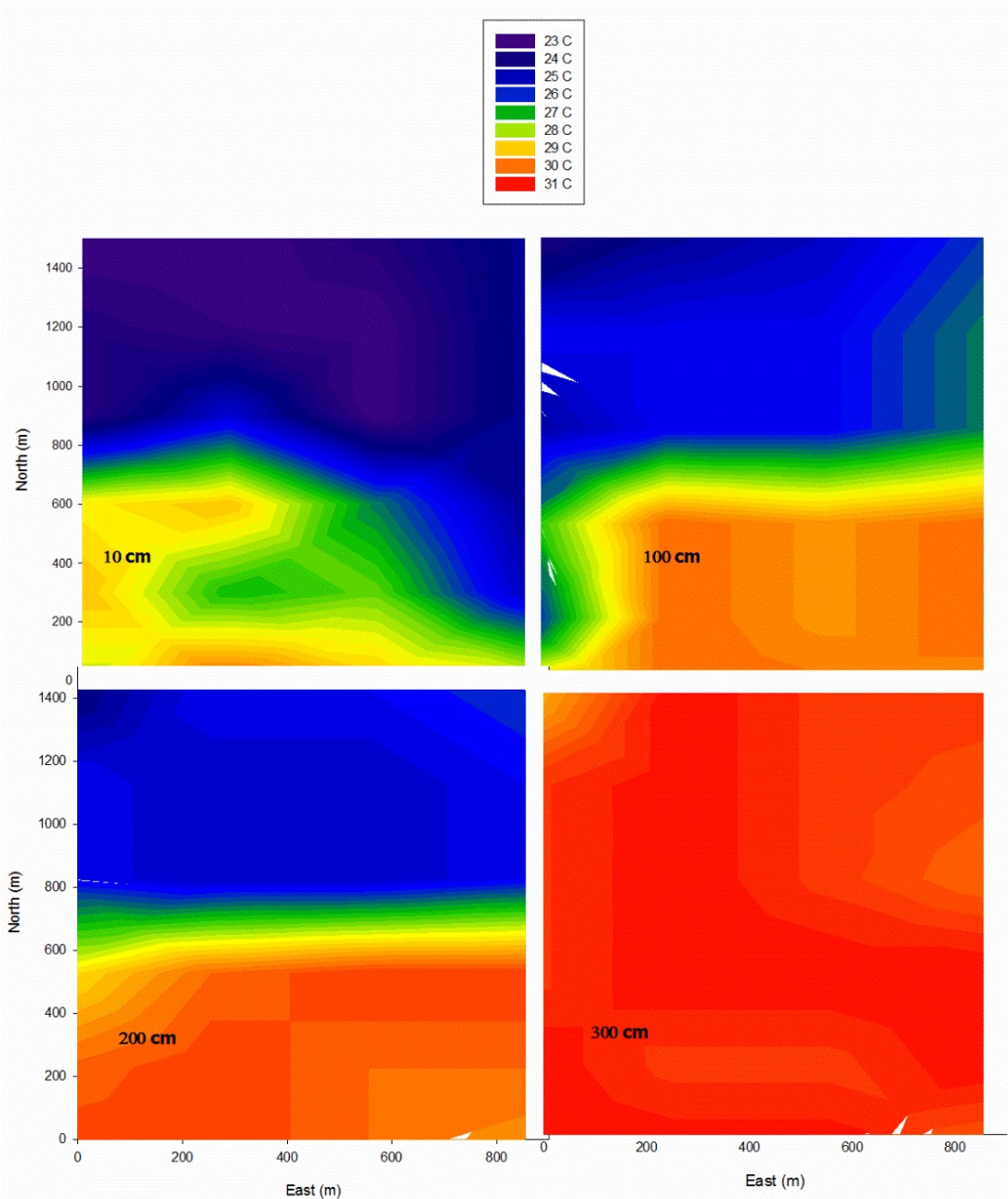

**Figure 11.** Air temperature at 0700 hour on 1 September at heights of 10, 100, 200 and 300 cm within the down gradient creosote bursage plant community. The solar facility was located approximately 100 m upgradient from the lower west to east edge of the monitoring area (see site location map).

*3.5. Soil Temperature*

Soil temperatures oscillated in a sinusoidal fashion with peak high (63.6 °C) and low (−5.0 °C) temperatures occurring at the 1 cm depth (tower DGC). The amplitude of the daily oscillations dampened with depth, with no statistical differences between average soil temperatures at all depths and between tower locations ($p > 0.05$). Only at the 10 cm depth at tower DGC did we observe greater variation over time compared to tower AF, yet there was still no statistical difference. Soil temperatures near the surface (1 cm depth) exceeded 40 °C during the day from May to October, associated with little response in soil volumetric water content at the 40 cm depth (Figure 3), suggesting that only heavy rains during this period (heavy evaporative demand) would potentially contribute to soil water recharge and deeper redistribution. Average air temperature based on the average of high and low temperatures on a monthly basis was 20.8 °C, which was the major driver of

the oscillation pattern around the average soil temperature with depth. Soil temperature during June, July and August (prime egg laying and egg maturation period of the desert tortoise) averaged 35.35 +/− 1.83 °C at the 40 cm depth at towers AF and DGC, indicating that burrow depth to achieve a sex ratio near 1.0 (31.8 °C, [38], 31–32 °C [39], with high mortality at 35.3 °C) would have to be at a depth significantly deeper than 40 cm at our site.

*3.6. Energy Balance*

We did not deploy eddy covariance towers but we did measure the incoming available energy as Net Radiation minus soil heat flux (Rn-G, 24 h, 365-day average) at three sites near the large-scale solar photovoltaic facility in Eldorado Valley. We assumed in a desert environment that had minimal precipitation that precipitation would be converted to latent heat of evaporation (mm to W/m$^2$). This assumed that minimal amounts of water moved beyond the rooting depth of creosote. Soil volumetric water contents typically revealed little change at depths of 40 cm (Figure 3) near the towers and only on a few occasions did we measure elevated soil volumetric water content at a depth of 100 cm inside the wash system. We selected only 2017 for the energy balance closure because of low precipitation and minimal indication of water movement to a depth of 100 cm. Because we measured higher soil water contents at tower DGC compared to tower AF, yet precipitation totals were identical we included an 18% runoff contribution based on differences in cumulative storage change at tower DGC compared to tower AF but only during months which had rainfall events greater than 1 cm/day (17 rainfall events during 2016–2019). These adjusted latent heat of evaporation estimates were then subtracted from the Rn-G estimates to provide for a first order approximations of sensible heat. In Table 2 we report the different energy component estimates for the monitoring period, indicating that all sites were dominated by sensible heat. However when we looked at the return of longwave radiation (Table 3) we found that the yearly average hourly longwave return was only 3.2 to 6.3% higher at towers near the solar facility compared to the DGC tower (significant at $p < 0.001$), however when we focused on December which was the coldest month, return of longwave radiation was 10 to 14% higher at the two towers near the solar facility compared to the DGC tower ($p < 0.001$), suggesting that the solar panel arrays were altering the energy balance and the return of longwave radiation.

**Table 2.** Energy balance (W/m$^2$, hourly average) for Rn-G, LE and H by closure for Towers IF, AF and DGC for the year 2017. Where Rn is net radiation, G is soil heat flux, LE is latent heat of evaporation and H is sensible heat. Significance based on Kruskal-Wallis one-way ANOVA on ranks ($p < 0.05$). Values with different letters indicate significant differences at $p < 0.05$.

| 2017 | Rn-G | LE | H |
|---|---|---|---|
| Tower IF | 80.69 [a] | 0.27 [a] | 80.42 [a] |
| Tower AF | 90.84 [a] | 0.24 [a] | 90.60 [a] |
| Tower DGC | 88.12 [a] | 0.24 [a] | 87.88 [a] |

**Table 3.** Longwave radiation (watts/m$^2$, yearly hourly average and December hourly average in 2017) along with mid-afternoon albedo's estimated in July 2017. Significance based on Kruskal-Wallis one-way ANOVA on ranks ($p < 0.001$). Values with different letters indicate significant differences at $p < 0.05$.

| 2017 | Return Longwave Radiation, Yearly Hourly Average | Return Longwave Radiation, December Hourly Average | Albedo |
|---|---|---|---|
| Tower IF | −94.3 [a] | −89.3 [a] | 0.172 +/− 0.007 [b] |
| Tower AF | −91.5 [b] | −86.0 [b] | 0.155 +/− 0.004 [a] |
| Tower DGC | −88.7 [c] | −78.4 [c] | 0.168 +/− 0.008 [b] |

## 4. Discussion

Increased energy demands and a need to shift to cleaner energy has led to accelerated development of solar energy [5,11], especially in the southwestern U.S. where irradiance levels are high on a year-round basis. Such a decision however needs to be well thought out as such large-scale development does not come without some environmental consequences [12,40]. In the Mojave Desert, shrubland ecosystems home to the threatened desert tortoise are fragile and slow to recovery from disturbance and when one facility which is km$^2$ in size merges with the next, habitats become fragmented, including the hydrologic decoupling between up gradient and down gradient plant communities.

We are unaware of any prior studies investigating the impact of utility scale PV systems on adjacent desert ecosystems, where the soil–plant–water–atmospheric system was assessed. In particular the monitoring of leaf xylem water potential, chlorophyll index, the spatial monitoring of air temperature and the quantification of returning longwave radiation, with the intent of defining a zone of impact. Our research in Eldorado Valley NV suggests that a service road built over 27 years before the solar facility was constructed, decoupled the flow of water from up gradient washes to down gradient washes and once decoupled, altered the area in which rain water harvesting occurred. Schwinning et al. [41] argued that the overall health of desert ecosystems is directly linked to the integrity of their surfaces and such drainage systems. In 2016, associated with high rainfall (only at tower IF which was located inside the solar facility on the up gradient south side) led to significant runoff and erosion, under cutting structural support to a large number of panel arrays, suggesting that maintaining wash connectivity would benefit not only the down gradient plant community but the solar facility as well. If leaving the native wash system intact within a solar facility is opposed by engineers, constructing a below ground wash system to move up gradient runoff to down gradient washes should be investigated as an alternative way to maintain flow.

The decoupling of the wash system at our site led to a significant decline in soil moisture, canopy level NDVI values and mid-day leaf xylem water potentials. Measurements along a 900 m gradient suggested that plants within the first 300 m from where the wash was decoupled were placed under significantly greater stress during the higher environmental demand summer months. To what extent these plants in the first 300 m from the decoupling altered habitat suitability for the desert tortoise is unknown and worthy of further studies. Limited data collected from an adjacent site not down gradient from the solar facility but still under the influence of the road revealed non-significant differences (with tower AF values) in soil water content and leaf xylem water potentials on two separate occasions, suggesting that the hydrologic decoupling was due primarily to the service road (service roads are part of all solar installations). Although creosote and bursage are drought adapted [42]), the lack of runoff to compliment the low levels of precipitation over an extended period of time had a direct effect on the shrubs closest to the zone of decoupling. However, it should be noted that the plants did not visually appear to be stressed suggesting that these plants could lower internal water potentials (extreme anisohydric, [43]) to favor steeper gradients to access water at perhaps deeper depths and also at greater lateral distances (not assessed in this study). However, our interpretation was based on monitoring creosote for a two to four year period after a 27-year period of wash decoupling. Creosote is a long lived shrub with Bowers [44] reporting an average maximum longevity of 330 years in the Sonoran Desert, which led her to state that long extended periods are needed to assess longevity, mortality and recruitment associated with droughts. Likewise, a true assessment of the impact of utility scale solar facilities on desert ecosystems also needs long term monitoring.

Air temperatures measured at the three towers revealed warmer night time temperatures at the two towers located in close association with the solar facility (IF and AF), with warmest temperatures compared to tower DGC occurring during the hours prior to 0900. These findings are in general agreement with results reported by Barron-Gafford et al. [9] who reported nighttime temperatures over a photovoltaic plant regularly 3–4 °C

warmer than over wildlands, representing a heat island effect. It should be noted that we did not monitor meteorological parameters within the solar panel arrays as was reported by Broadbent et al. [10]. They found no nocturnal differences in air temperatures at the 1.5 m height when compared to their reference site, indicating that a nocturnal stable layer did not form below the 1.5 m height inside the panel arrays. We chose a height of 1 m to focus our assessment as it was the height of the vegetative cover. Within panel arrays Broadbent et al. [10] reported a decrease in nocturnal soil cooling because of downward transport of sensible heat and reduced net longwave radiation from the soil associated with a lowered ground sky view factor with the photovoltaic modules. We believe it is possible that sensible heat trapped below panels may have been displaced by cold air drainage during the night with greatest differences between towers adjacent to the solar facility and tower DGC occurring in the early morning hours.

Although we measured fewer freezing temperatures in the area near the solar facility which would benefit the plants, higher night time temperatures would also be associated with elevated night time respiration which would represent a cost to the plants [45,46], perhaps leading to a slow long-term decline in the overall health of the plants such as through decreased membrane thermal stability as reported in rice [46], a subject worthy of further investigation. Air temperature measurements in an adjacent down gradient location from the solar facility revealed significantly warmer nighttime temperatures on a larger spatial scale. As the warmer air was displaced down gradient, the temperature front advanced into the creosote—bursage plant community with values 5 to 8 °C warmer along the southern edge of the grid at the 1 m height. However, this front was more extensive at the 3 m height reflecting the convection of the warmer air [47]. The data suggested that significant heat was moving from the solar facility into the plant community, especially in the first 200–400 m, overlapping with the wash decoupling zone. Determining the biological significance of these elevated temperatures will require further research. The solar panel arrays in the solar facility associated with this study were a fixed panel system. However, if the panels had been mounted as a tracking system, the panels could have been situated in a perpendicular position relative to the ground at night allowing longwave radiation and trapped sensible heat [9] to escape to the sky, reducing the heat displacement into the adjacent plant community during the early morning hours (research needed). However, Broadbent et al. [10] argue that more efficient transport of sensible heat occurs when the panels are tilted. Different systems now exist in Eldorado Valley that a side-by-side comparison of heat transport associated with different panel arrangements could be made. We believe a combination of sensible heat, advection, topography, wind direction and cold air drainage interact in complex ways to influence heat movement away from solar facilities in Eldorado Valley. However, we recognize the limitations of our experimental approach; short monitoring period relative to the life expectancy of creosote plants and the PV solar facility, not monitoring inside the panel arrays and not incorporating a wildlife monitoring program.

## 5. Concluding Remarks

Fragmentation of desert ecosystems can be expected with large scale solar energy development. This fragmentation will be exacerbated by high-density placement of these facilities, which can be anticipated based on the investment in grid infrastructure in a given area. The key will be to maintain conservation value of these fragmented remnants [28] and to not obstruct corridor use [5]. Based on our research in Eldorado Valley, NV, USA a down gradient zone of about 300 m (footprint) was impacted to the greatest extent, suggesting that the spacing between solar facilities (policy decision) will be a critical factor in terms of preserving high quality habitat for the desert tortoise and other threatened species. Knowledge of what the minimum critical habitat size needed for such species is not fully known. Greater research is needed to identify habitat zones acceptable for organisms within areas of high solar energy development and this should be done prior to any fragmentation of the ecosystem. Long-term monitoring of utility scaled PV facilities

built within desert ecosystems is needed to clearly capture the response on a timescale that has greater meaning relative to the life expectancy of the plants and animals within these ecosystems. Finally, we agree with Hernandez et al. [30] that solar energy development should merge engineering solutions with biological solutions to achieve long term energy sustainability.

**Author Contributions:** Conceptualization, D.A.D.; Methodology, D.A.D. and B.B.; Formal Analysis, D.A.D., L.A., B.B., L.F., J.P.D.J. and M.D.P.; Writing, D.A.D.; Review, editing D.A.D., L.F. and M.D.P. All authors have read and agreed to the published version of the manuscript.

**Funding:** National Science Foundation—EPSCoR 11A-1301726.

**Informed Consent Statement:** Not applicable.

**Data Availability Statement:** Contact corresponding author.

**Acknowledgments:** This research was supported with NSF EPSCoR funding to D.A. Devitt. The authors also wish to acknowledge the contribution of Cheryl Collins of the Desert Research Institute for quantifying percent cover from extracted Google images.

**Conflicts of Interest:** The authors declare no conflict of interest.

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
