# Peer review of "Assessing the Impact of a Utility Scale Solar Photovoltaic Facility on a Down Gradient Mojave Desert Ecosystem"

_land, doi:10.3390/land11081315_

Round 1
Reviewer 1 Report
The work is very interesting and, in my point of view, innovative.
My suggestion is to improve the readibility of figure 1 and 5.
Author Response
Response to Reviewer #1
1) Provide more relevant references. We added 12 new references of which 8 were published between 2018 and 2021.
2) Improve readability of figures 1 and 2.
We added a new figure 1 which reports the soil volumetric water content data as a box plot which reveals the median value, upper and lower quartiles and potential outliers. We decided to also keep the old figure 1 and rename it figure 2 as we believe the raw data shown int the new figure 2 reveals the episodic nature of the soil water content and the larger number of observations above a soil water content of 0.10 which we felt was a clear indication of recent rainfall events. Figure 5 which is now figure 6 we added time brackets and inserted the median values and p values to distinguish statistical differences between the two sites over the period reported.
Reviewer 2 Report
Overview:
The manuscript with the title " Assessing the Impact of a Utility-Scale Solar Photovoltaic Facility on a Down Gradient Mojave Desert Ecosystem " aim to assess the influence of a large photo voltaic facility on heat and water transport into an adjacent creosote (Larrea tridentata) bursage (Ambrosia dumosa) plant community.
However, there are various things that need to be adjusted in the revised version.
Detailed comments:
Introduction:
- The literature is not reviewed well; the authors must need to review the recently published articles. There are multiple recent studies (From 2019 to 2021) that also need to be cited in the revised version. The authors should clearly show what have we done and why this study is important in the introduction section.
- I couldn't find the innovation of this research, what is the research gap and why this study is important? Please elaborate
- There must be a space prior to each reference.
Material and Results:
- Why the authors have used Satellite NDVI values? Please check this.
- Line 167: Data was processed using Sigma Stat and graphs generated using SigmaPlot (Systat 167 Software Inc. Point Richmond, CA), how the authors think that this is the correct representation of data and research, please elaborate.
- Did the authors consider the weather circumstances in each area when they have calculated the PV annual power generation?
- Why does the analysis use 2015-2016 data why not the latest data of 2021?
6. The authors need to provide more details on the sampling and data collection.
7. What are the limitations of this study?
8. what are the conclusions of this research, authors should write at the end of the manuscript.
Best,
Author Response
Literature not reviewed well - we added 12 new citations of which 8 were from 2018-2021
What is the research gap - in the last paragraph of the introduction we begin with defining the research gap.
space prior to each reference - we have inserted a space after each comma separating references.
Why use Landsat NDVI values to compare with plant level NDVI sensor values ( now stated , p>0.05), demonstrating that the satellite data could not detect the subtle changes reported at the plant canopy level (figure 6). Although large areas can be quickly assessed with satellites , care should be taken in drawing conclusions about the impact utility scale PV systems have on desert ecosystems with low resolution satellite data - (cite Potter 2016).
How authors think this is the correct representation of the data using Sigma Stat. SigmaStat reports that over 200,000 scientists and engineers have used SigmaStat worldwide and that SigmaPlot has been used in the scientific community for over 30 years. I personally have used SigmaStat for over 20 years reporting statistical analysis in over 40 peer reviewed publications. SigmaStat includes software for ANOVA , ANCOVA, PCA and AIC to mention just a few. Finally we have confided with our statistician who approves the use of SigmaStat.
Did the authors consider the weather circumstances. Yes, we monitored ETref, precipitation, Rn-G and night time long wave radiation at each site, along with air temperature on a large spatial scale down gradient from the PV facility. We utilized all of this data to describe water and heat flow, however, we did not calculate PV annual power generation.
Why does the analysis use 2015-2016 data and not the latest data of 2021.
We explained in our cover letter to the editor that the PhD student on this study monitored the experiment in 2015 and 2016, passed his PhD written and oral exams and then requested a one year leave of absence and then decided to leave the program all together. It took over a year to retrieve his data files. As such the soil plant measurements only occurred in 2015 and 2016, whereas the meteorological towers continued to provide data through 2020.
More details about sampling and data collection. We read over the M&M many times looking for how we could provide a more detailed description of the sampling and data collection. we did add a new citation for the NDVI sensors but otherwise we were not successful in addressing this concern. We would be happy to address any specific comments the reviewer might have.
What are the limitations of this study. We added several sentences in the Discussion section that clearly state the major limitation of this study. "However, we recognize the limitations of our experimental approach ; short monitoring period relative to the life expectancy of creosote plants and the PV solar facility, not monitoring inside the panel arrays and not incorporating a wildlife monitoring program"
What are the conclusions of this research? We modified the final paragraph and relabeled it "Concluding Remarks". The final concluding sentence states that "we agree with Hernandez et al.[30] that solar energy development should merge engineering solutions with biological solutions to achieve long term energy sustainability.
Reviewer 3 Report
Abstract
The abstract of this paper provides a thorough overview of the main components of the same, by talking about, the influence that a large sized photovoltaic facility can have on water and heat transport, in an adjacent and bursage plant community. The purpose of the paper, including the rationale behind the study, the fact, that it has been conducted in the context of the Mojave Desert in the USA, and the concluding arguments which is that the development of high solar energy is necessary is necessary prior to ecosystem fragmentation, has been made clear in the abstract itself.
Introduction
The introductory section of this research paper is one that is very detailed, and it gives a concrete overview of the context of the study, including the range and type of solar development that has been taking place in the American state of Nevada, up until now. Why the Mojave Desert has been chosen as the field site, is justified in the introductory section itself, with the main scope of the study being pointed out as well, which is to understand how plants beneath the desert surface can be impacted because of the development of high solar energy, in the Mojave Desert.
Literature Review
A lot of the literature that pertains to the undertaking of excessive solar energy development in the USA has been reviewed in the introductory section itself and there is no separate section of the paper that has been demarcated for the literature review only. Key studies that have been conducted on why it is that solar energy development is so fruitful in the desert regions of the USA, such as in the state of Nevada, is what is highlighted in the review of literature and the justification for the current study has been quite aptly, put forward. \
Methods
The materials as well as the methods that have been used for carrying out this study has been put forward quite clearly in this paper as well. That the Eldorado Valley in the Mojave Desert region of Nevada was chosen to be the site of study, is made known, and the plant transects that were established for the study has been pointed out with clarity here as well.
Results
The data which was collected for this study has been analyzed and then presented in the form of tables, and charts, with the main argument being that high intensity solar energy development ought not to be undertaken in the desert region in Nevada as this is capable of having an adverse impact on the plant life that is growing beneath the desert surface. The role that the soil of the region can have in influencing such an impact is also elaborated upon in the results section.
Conclusion
The study concludes that there is a negative effective associated with the impact of ultraviolent and photovoltaic facility on water and heat transportation in adjacent desert cresol.
Comments
I go through a well-written article with an intriguing topic and a typical research technique, however the work has inadequate literature and references. It is necessary for the references to enhance the significance. In addition, the sections of Discussion and Conclusion need to make it abundantly obvious what the contribution of the research is, as well as any pertinent policy ideas, originality, limits, and recommendations for further study. In the future, there is further in-depth research of a similar sort that has to be carried out. This is necessary in order to provide more definitive conclusions in the context of this study.
Author Response
Inadequate references - we added 12 new references of which 8 were published between the years 2018-2021.
Rework the discussion and conclusion section.
We added several sentences that clearly indicate the originality of the study. we also added several sentences to spelling out the limitations of the study. Finally we converted the last paragraph into "Concluding remarks" with a definitive conclusion that "we agree with Hernandez et al. [30] that solar energy development should merge engineering solutions with biological solutions to achieve long term energy sustainability"
Round 2
Reviewer 2 Report
It is considered that the new version presented by the authors gives an adequate answer to the main questions previously presented.
Author Response
Thank you for your valuable comments.